# Milk Whey Protein Fibrils—Effect of Stirring and Heating Time

**DOI:** 10.3390/foods13030466

**Published:** 2024-02-01

**Authors:** Gunvantsinh Rathod, Jayendra Amamcharla

**Affiliations:** 1Department of Animal Sciences and Industry, Food Science Institute, Kansas State University, Manhattan, KS 66506, USA; gunvantsinhrathod@ksu.edu; 2Idaho Milk Products, Jerome, ID 83338, USA; 3Midwest Dairy Foods Research Center, University of Minnesota, St. Paul, MN 55108, USA

**Keywords:** milk whey protein fibrils, fibril formation, stirring and static heating, viscosity

## Abstract

Milk whey proteins, which are derived from skim milk through membrane filtration, exhibit valuable functional properties when transformed into a fibrillar form. This conversion enhances their suitability for various applications, including thickening, gelling, emulsification, and foaming. However, reported fibrillation methods have longer heating times, which may not be economical for the dairy industry. To address these challenges, the current study was undertaken with the objective of reducing the time required for fibril formation. In this study, 2% milk whey protein isolate (mWPI) solution at pH 2 was heated with static and stirring heating conditions at 80 °C for 20 h to convert milk whey proteins into fibrils. Fibrils were observed using the thioflavin T value, transmission electron microscopy, Tricine SDS-PAGE, rheology, and protein oxidation. Results suggest that stirring heating conditions with 14 h heating time produced fibrils with good morphology compared to static heating, showing a 6 h reduction compared to an earlier reported 80 °C for 20 h heating time. Also, stirring heating produced a uniform and homogeneous fibril solution compared to the static heating method. Gentle stirring during heating can also help to scale up fibril production in an industrial setup. The fibrillation method with processing intervention will help to produce fibrils with enhanced functionality at the pilot and industrial scales.

## 1. Introduction

Milk proteins are an excellent source of nutrition and provide a range of essential amino acids. In addition to nutrition, globular milk proteins such as whey proteins provide excellent functional properties such as thickening, gelling, emulsification, foaming, and surface activity. These functional properties can be further enhanced with different physical, chemical, and enzymatic processing interventions [1,2]. These functional properties can be further enhanced by altering the globular structure of the protein to a fibrillar structure [3,4] using the fibrillation process. Structurally, these fibrillar proteins are a few nanometers in diameter and a few microns in length and have a high aspect ratio, which can provide a space-filling ability to fibrils [5]. Also, due to the fibrillar structure, fibrillar proteins can provide the previously mentioned properties at a relatively lower protein content than the native proteins. The fibril formation process of different plant-based food proteins such as soy protein [4], pea protein [6], maize protein [7], wheat protein [8], rice protein [9], and cottonseed protein [10] have been extensively studied. Other than plant proteins, animal proteins such as egg and milk proteins were also explored for fibril formation [3].

Milk proteins are composed of two principal proteins casein and whey proteins, among which whey proteins, a globular protein, were extensively studied for fibril formation. There were several studies on the fibril formation of whey protein isolate (made from cheese whey) and a pure β-lactoglobulin (β-Lg) fraction of whey proteins [11]. These studies were conducted at different concentrations of whey proteins (1–3%), with a varying pH of 1.8 to 3 (below the isoelectric point of whey proteins) and different time–temperature combinations from 75 to 120 °C and 5 to 24 h [3]. Among them, fibrils formed by heating (at 80 °C) 2% whey protein solution (at pH 2) for 20 h was reported to be the best among all other combinations in terms of fibril morphology (intertwined, straight, semiflexible, etc.), length (up to several microns), and the maturity of fibrils [12]. During the fibrillation process, peptides self-assemble and form protofilament structures, which undergo structural rearrangements until they reach a final stable fibril structure, called matured fibrils [13]. However, 20 h is a very time-consuming process and is deemed not practical for the industrial-scale production of whey protein fibrils. Therefore, a faster method is required to make fibrils without compromising the quality of the fibrils. Earlier studies reported the use of higher temperatures (85–95 °C) to shorten the fibril formation time, which can produce fibrils; however, the morphology of fibrils (in terms of fibril length, diameter, and uniformity of fibril structure) was not identical to that of fibrils produced at 80 °C. Similar results were found when different time–temperature combinations such as 80 °C for 20 h, 85 °C for 5 h, 90 °C for 6 h, and 95 °C for 6 h were carried out in our preliminary studies for the fibrillation of milk whey protein isolate (mWPI) solution (Figure 1). The whey protein isolate produced using microfiltration and ultrafiltration of skim milk is generally referred to as mWPI, as it does not contain glycomacropeptide, a C-terminal part of κ-casein. During fibril formation, proteins are hydrolyzed into smaller molecular protein fractions (2–8 kDa) due to heat and low pH. Among hydrolyzed protein fractions, some hydrolyzed proteins act as nuclei and other protein fractions arrange themselves in a systematic order to form fibrils [14,15]. Hence, hydrolysis of proteins and the subsequent nuclei growth in terms of fibril formation is most important to make the fibril formation process faster. Most of the reported study was conducted using static heating in a glass container, where the movement of the hydrolyzed peptides (formed due to heat and low pH) is lower, hence the growth of nuclei takes time to form fibrils, followed by the uniform growth of nuclei into mature fibrils. Considering the importance of nuclei formation and the availability of hydrolysate for fibril formation, the intervention of adding preformed fibrils as seeds and stirring the pre-acidified protein solution with added seed fibrils while heating was tried [16]. They reported that seeding did not affect fibril growth; however, the stirring of acidified protein solution can increase the overall rate of reaction and accelerate the fibril growth by enhancing the material flow in the solution. However, the reported studies did not show the optimum time–temperature combination to produce highly functional fibrils in terms of their morphology, maturity, and functionality. Therefore, this study was designed to understand the effect of stirring and no stirring with an aim to minimize the fibrillation time without compromising the quality and morphology of the fibrils.

## 2. Materials and Methods

### 2.1. Experimental Approach

Two lots of mWPI (4.3% moisture, 91.6% true protein, 1.4% lactose, and 2.1% ash) were procured from a commercial manufacturer. mWPI from each lot was reconstituted and subjected to two different fibrillation processes, viz., static (control and no stirring during heating) and stirring during heating, to compare fibril formation with regards to the fibrils’ structure, morphology, conversion from native protein to fibrils, rheology of fibrils, and protein oxidation to produce good quality fibrils with a reduced fibrillation time. The detailed protocol is explained in subsequent sections. The chemicals used in the analysis were of analytical grade. 

### 2.2. Process for mWPI Fibril Formation

The method provided by Rathod and Amamcharla [17] was used to produce mWPI fibrils. Briefly, mWPI powders were rehydrated to 2% (*w*/*w*) solution on a protein basis with distilled water and stored overnight at 4 °C to ensure complete hydration. The pH of mWPI solution was adjusted to 2 using 6 N hydrochloric acid, and the solution was stored overnight at 4 °C. The pH-adjusted mWPI solutions were heated to 80 °C for 20 h using two different approaches. In approach 1, the mWPI solution was transferred into 20 heat-resistant, screw-cap test tubes (8 mL, 17 mm dia. × 63 mm height) (DWK Life Science, Millville, NJ, USA), placed in a GPD 20 water bath (Fisher Scientific, Waltham, MA, USA), and maintained at 80 ± 0.5 °C. Every hour (starting from 2 h), one tube was taken out without disturbing the other tubes. The mWPI solution was cooled to 4 °C by placing the tube in an ice water bath. In approach 2, the mWPI solution (500 mL) was heated (at 80 ± 0.5 °C) in a conical flask using a temperature-controlled circulatory water bath (Cole Parmer, Vernon Hills, IL, USA) under gentle continuous stirring using a magnetic stirrer. The conical flask was sealed to ensure no moisture loss during heating. A thermocouple (Omega Engineering Inc Model: RDXL4SD 4 Channel, Norwalk, CT, USA) was used to continuously monitor and record the temperature during the heating process. Every hour (starting from 2 h), a representative sample (10 mL) was quickly drawn using a pipette and transferred into the glass tube, with subsequent cooling and labeling similar to approach 1. Care was taken to quickly draw the sample to avoid unwanted moisture loss from the remaining solution. Samples were stored in the refrigerator at 4 °C till further analysis. All the samples were analyzed for thioflavin T fluorescence value, transmission electron microscopy, gel electrophoresis, rheology, and protein oxidation.

### 2.3. Confirmation of the Presence of mWPI Fibrils

#### 2.3.1. Thioflavin T (Th T) Fluorescence Value 

The Th T value was measured using the method described by Rathod and Amamcharla [17]. Briefly, 48 μL of mWPI fibril solution was mixed with Th T solution and analyzed with a spectrofluorometer at excitation and emission wavelengths of 440 nm and 482 nm, respectively. Each sample was tested in duplicate, and the average of the two lots was reported. 

#### 2.3.2. Transmission Electron Microscopy (TEM) 

Briefly, mWPI fibril samples were diluted to 0.05% protein using distilled water at pH 2. TEM was performed using the method described by Rathod and Amamcharla [17]. 

#### 2.3.3. Tricine Sodium Dodecyl Sulfate-Polyacrylamide Gel Electrophoresis (SDS-PAGE) 

mWPI fibril solutions were diluted to 0.5% protein from the initial 2% protein solution. Then, samples were diluted with Tris–Tricine sample buffer in a 1:2 ratio and mixed using a vortex mixture, followed by incubation at 37 °C for 15 min in a water bath. Then, 15 μL of the diluted sample with buffer was loaded in the 16.5% precast Tris–Tricine gel (Bio-Rad Laboratories, Hercules, CA, USA) and placed in a gel electrophoresis assembly (BioRad Mini Gel system, Bio-Rad Laboratories, Hercules, CA, USA) filled with 1X Tris–Tricine sample buffer (Bio-Rad Laboratories, Hercules, CA, USA). The gel was run on 20 mA for 15 min for stacking. Then, it was adjusted to 25 mA for 90 min. Staining, destaining, and image processing was performed as suggested by Rathod et al. [18]. Images were taken with a camera and analyzed with ImageJ software Version 1.53f (ImageJ, National Institutes of Health, Bethesda, MD, USA). 

### 2.4. Rheology

Rheology of mWPI fibril samples was performed at 20 °C, as per the method described by Rathod and Amamcharla [17], using a stress-strain-controlled rheometer (MCR-92 Anton Paar, Vernon Hills, IL, USA) fitted with a 50 mm diameter stainless steel cone with an angle of 1° and 101 μm gap at a varied shear rate from 0.01 s^−1^ to 200 s^−1^. Apparent viscosity was recorded at 100 s^−1^. Each sample was tested in duplicate, and the average of the two lots was reported.

### 2.5. Protein Oxidation

Protein oxidation was studied according to the method given by Scheidegger et al. [19] and Keppler et al. [20]. Briefly, mWPI fibril samples were diluted to 0.1% protein using distilled water at pH 2. Protein oxidation was estimated in terms of decay in two amino acids, viz., tryptophan and tyrosine, and their respective increase in oxidation products, which are N-formyl kynurenine and Di-tyrosine. Diluted protein solutions were analyzed in a quartz cuvette using a spectrofluorometer (LS-55; Perkin Elmer, Waltham, MA, USA) using a right-angle assembly for tryptophan (at 294 nm excitation and 340 nm emission) and L-tyrosine (at 274 nm excitation and 310 nm emission). Similarly, oxidation products, N-formyl kynurenine (at 325 nm excitation and 435 nm emission) and Di-tyrosine (at 284 nm excitation and 415 nm emission), were measured as suggested by Rathod et al. [18]. The fluorescence intensity was plotted against time as an AU (arbitrary unit) to see an increase or decrease in the above-said component with respect to time. Each sample was tested in duplicate, and the average of the two lots was reported.

### 2.6. Statistical Analysis

Statistical analysis was performed by repeated measures ANOVA using SAS Version 9.4 (SAS Institute Inc., Cary, NC, USA). The experiment was replicated twice with two lots of mWPI. Averages and standard deviations were calculated using Excel (Microsoft Corp., Redmond, WA, USA).

## 3. Results and Discussion

### 3.1. Confirmation of the Presence of mWPI Fibrils 

Fibril formation was confirmed using the Th T value, TEM, and Tricine-SDS PAGE. 

#### 3.1.1. Th T Value

The Th T values of the static heating and stirring heating methods are shown in Figure 2. The Th T values of unheated mWPI solution (control) at pH 2 were 35.28 ± 4.05 and 32.63 ± 3.78 for static and stirring heating samples, respectively, which were considered the control or reference Th T values. With the increase in time of heating from zero hours, the Th T value increased for both static and stirring heating; however, the increase in Th T value was not the same (Figure 2). At 10 h, the Th T value of static heating was 277.88 ± 58.8 AU, which was significantly (*p* < 0.05) lower than that of stirring heating (417.65 ± 48.09 AU). Further, at 20 h, the Th T value of static heating samples was 404.27 ± 25.05 AU, which was significantly lower than stirring heating samples (683.07 ± 90.62 AU) and even lower than the Th T value of the stirring heating sample at 10 h (417.65 ± 48.09 AU). Further, the Th T values of the stirring and static heating samples were significantly (*p* < 0.05) different from each other during the whole 20 h heating duration, and stirring heating showed faster fibrillation than static heating (Figure 2). 

The Th T value is an indicator of fibrillation which increases with an increase in fibril formation [11]. The result showed that stirring heating had a higher Th T value than static heating during the 20 h of heating, showing significantly higher fibril formation with stirring heating compared to static heating. Loveday et al. [21] also observed a similar rise in Th T value when β-Lg solution (pH 2) was heated to 80 °C in a glass vial over a period of 24 h. Further, they divided fibril growth into an initial slow increase denoted by the lag phase, a rapid increase denoted by the log phase, and little rise denoted by the stationary phase. To find the fibril growth, the graph of the Th T value against time (Figure 2) was further analyzed for slope for different time durations up to a total of 20 h. It was observed that, till 3 h, the slope value for static heating and stirring heating was 38.8 and 20.6, respectively, followed by a slope value increase to 43.5 and 26.3, respectively for the 3–10 h duration, and then the slope value was reduced to 37.5 and 21.8, respectively for the 10–20 h duration. Therefore, it can be said that fibril formation in the current study also followed a similar lag phase, log phase, and stationary phase. However, at each stage, the stirring method of heating showed a higher slope value denoting faster fibril formation. For fibril formation, the mWPI solution was heated at pH 2 in both methods. Heating of the protein solution at a low pH facilitates protein unfolding and faster cleavage of peptide bonds, resulting in unfolded protein molecules and peptides, which are prone to rearrange to make a fibrillar form of protein at a low pH [3]. During the fibrillation process, these peptides act as nuclei and grow as fibrillar structures. In static heating conditions, these rearrangements and fibril formations will be performed with unfolded proteins and peptides available near the nuclei due to the limited movement of the protein solution. Therefore, fibril formation will be slow once unfolded proteins and peptides are depleted from the solution, and the resultant fibril will be formed from available peptides and protein fractions. In the case of stirring heating, rearrangement and fibril formation will be faster due to faster movement of the liquid, providing appropriate protein fraction for fibril growth. Therefore, the stirring heating method showed faster fibril formation. Bolder et al. [16] reported that continuous stirring may break up immature fibrils and produce more mature fibrils. 

#### 3.1.2. TEM

TEM images of fibrils formed under static and stirring heating conditions at different time intervals are shown in Figure 3. These TEM images show fibril formation from the native globular whey proteins to fibrillar form in both the heating conditions starting from 2 h. TEM images of fibrils are presented at two magnifications, where 3400× was observed to see the overall arrangement and aggregation, while 34,000× was observed for fibril morphology, diameter, length, and arrangement, such as separate or entangled, intact or fragmented. Other higher magnification (92,000×) images were used to measure the dimensions of the fibrils. Under static heating conditions, very few visual fibrils can be seen in the 2 h sample. Fibril formation can be seen, but it is not as intensive as during the stirring heating. Large aggregates and fibrillar aggregates can be seen in the static heating. Unlike static heating, stirring heating showed a uniformly distributed network of fibrils, and the visible fibrils were higher as compared to static heating. Relating this observation to Th T value results, stirring heating showed a higher Th T value than static heating. With an increase in heating time, fibril formation can be seen in TEM images of both methods, but stirring heating showed more visible and uniform fibrils compared to static heating. Also, visible aggregates were less in the stirring heating condition as compared to static heating. Th T value results also suggested that stirring heating showed a higher Th T value with an increase in heating time, and the same increase in fibril formation can be seen in the TEM images. At 14 h, fibrils from stirring heating were clear, intact, and mature, which can be seen in the image at lower magnification (overall fibril arrangement and very little aggregation) and higher magnification (clear, distinct fibril morphology) (Figure 3). Bolder et al. [16] reported that stirring accelerates the kinetics of fibril formation, resulting in an increase in the number of visible fibrils which can be seen with TEM images. However, further heating leads to a decrease in visible fibrils (Figure 3), as prolonged heating might break the fibrils and terminate the growth of fibrils [16]. Similarly, we can see an increase in the Th T value till 14 h and then the increase in the Th T value is smaller, which also suggests that heating beyond 14 h is not positively helping fibril formation. So, both results suggest limiting heat treatment to 14 h for better fibril formation. 

In the case of stirring heating, continuous movement of the solution facilitates the uniform heating of the mWPI solution, which results in uniform protein hydrolysis and production of peptides. Continuous stirring also facilitated the breakage of aggregates that were usually formed when the protein solution was heated [16]. Further, during fibril formation, stirring of the solution provides enough hydrolyzed protein/peptides to fibril nuclei to form fibrils. Hence, fibrils formed in the stirring heating condition were uniform and identical, showing less fibrillar aggregation as compared to static heating. Akkermans et al. [22] also reported that shear flow enhances fibril formation in heat-denatured β-Lg samples by facilitating the movement of protein solution and breaking earlier-formed, immature fibrils. In static heating, fibrils were formed based on the availability of hydrolyzed protein/peptides surrounding the fibril nuclei in the solution, and their amount cannot be even, due to a lack of movement of the solution, which occurs in the stirring heating condition. This results in the formation of non-uniform fibrils, which can be seen in Figure 3. Further, without any shearing action in the static heating condition, immature fibrils may not split and re-form, resulting in fibrils with an irregular thickness, which can be seen in higher magnification images. Further, the restriction in movement of fibril precursors also promotes the interaction and crosslinking of protein hydrolysates, resulting in the formation of aggregates. These aggregates can also be seen in the TEM images (Figure 3). Along with aggregates, gel formation due to crosslinking further limits the movement of hydrolysate and depletes the source of hydrolysate, a building block for fibrils. Similar fibrillar aggregates can be seen in the earlier reported studies where fibrillation was performed in static conditions [14,23,24]. Therefore, stirring heating can produce uniform fibrils. 

#### 3.1.3. Tricine SDS-PAGE

Tris–Tricine SDS-PAGE images are shown in Figure 4. Initially, both static and stirring heating methods showed clear bands of different whey protein fractions in their unheated whey protein sample at zero hours, among which α-Lactalbumin (α-La) and β-Lg are the major whey protein fractions, and their position was marked with respect to marker bands on the ladder (Figure 4A,B, lane L) based on their molecular weights, 14.2 and 18.4 kDa, respectively [25]. Other than major whey proteins (α-La and β-Lg), a small portion of other milk proteins with a higher molecular weight (>25 kDa) can be seen in the band of unheated whey protein solution (Figure 4A,B). However, there was no protein band less than 10 kDa. Upon heating whey protein solutions at pH 2, the thickness of whey protein bands was reduced with an increase in heating time from 0 to 20 h, and bands of small molecular weight fractions below 10 kDa were increased with heating times from 0 to 20 h. The gel image (Figure 4) showed that compared to stirring heating (Figure 4B), static heating (Figure 4A) showed thicker bands of small protein fractions (2–10 kDa) from the 2 h itself and showed a thinner band of intact whey proteins at the end of 20 h, which suggest a faster conversion of intact whey protein to smaller molecular weight peptides. Further, with an increase in heating time, bands of whey proteins that have more than a 25 kDa molecular weight also disappeared within a few hours of heating in both heating methods. This observation suggests that not only α-La and β-Lg but other whey protein fractions are also hydrolyzed during heating at 80 °C and pH 2, and they might contribute to the fibril formation.

In the fibril formation process, proteins were hydrolyzed into smaller-sized protein fractions of 2–8 kDa and rearranged into fibrils [14]. Under the action of SDS, fibrils which are made from small-sized protein fractions were dissociated again and seen as separate gel bands of respective size. Hence, the occurrence of a higher amount of small-sized protein (2–8 kDa) could give an idea about a higher rate of hydrolysis, which could enhance the formation of the fibril. But along with the number of small-sized protein fractions, systematic self-assembling was also needed for fibril formation. Stirring heating showed lower conversion of small-sized protein fractions, but the stirring heating condition may have enhanced movement and rearrangement of these small protein fractions and, thereby, fibril formation. This observation can be confirmed with TEM images of fibrils (Figure 3) and Th T values of fibril samples (Figure 2). Bolder et al. [26] reported that the β-Lg, not the α-La, was responsible for fibril formation. However, in the current study, we can see a reduction in band intensity of every intact protein band including α-La and other higher molecular weight (>25 kDa) proteins. So, this observation suggests that almost all the proteins were being hydrolyzed in the fibril formation process. However, whether peptide fractions from proteins other than β-Lg are contributing to fibril formation or not is still unknown, but the possibility of their contribution cannot be denied with this observation. 

### 3.2. Rheology

Table 1 shows the apparent viscosity (AV) of the fibril solution prepared with static and stirring heating conditions at different time intervals from 0 to 20 h. Here, the sample was subjected to a shear ramp (0–200 s^−1^), and the AV was recorded at a shear rate of 100 s^−1^. The AV (at 100 s^−1^) of the stirring heating sample increased from 1.08 ± 0.03 to 1.37 ± 0.19 mPa s within 2 h and increased continuously until it reached 6.08 ± 0.09 mPa s at 19 h. Unlike the continuous increase in the AV of fibril samples of stirring heating, fibril samples of static heating showed a definite increase in viscosity, but the increase was irregular, finally reaching 6.48 ± 1.10 at 19 h. Here, it is of note that stirring heating samples are more homogeneous and uniform, while static heating samples are inhomogeneous with some aggregates and particulates. Due to the presence of particulates and aggregates, rheology analysis was affected when they came into the narrow space between cup and bob, showing greater resistance to flow and, thereby, a high AV. Similar to the AV, an increase was also seen for the consistency coefficient, where the fibril sample of stirring heating showed a steady increase compared to the discontinuous increase with the fibril sample of static heating. In a similar pattern, the flow behavior index also decreased with time in both methods, but the fibril sample of stirring heating showed a gradual decrease compared to a discontinuous decrease in static heating. The probable reason could be the method of manufacture of the fibril. In stirring heating, pH-adjusted protein solution was continuously stirred, which facilitated the even distribution of hydrolyzed protein fraction for fibril formation; hence, the fibrils formed, aligned themselves in a similar orientation and remained parallel to each other in the solution. So, when they were analyzed for rheology, they may have aligned parallel to the plane of the probe. In addition, both samples showed shear thinning behavior; therefore, continuous stirring in the stirring heating method should have resulted in shear during fibril production itself, resulting in a lower AV value. However, their flow behavior index did not fall below 0.6 compared to the ~0.3 of static heating, showing that the stirring heating sample has a lower initial viscosity but retains viscosity better upon shear as compared to static heating samples. At a higher shear rate both the heating methods showed non-significant differences in AV. Further, the fibril solution of the stirring heating sample was clear and uniform without any particulate/aggregates; however, static heating samples showed visible particulate/aggregates, which could be the reason for an uneven increase in the AV of static heating samples. Static heating fibrils may have formed a three-dimensional network and localized aggregation during the fibril formation process as the protein solution remained undisturbed during heating [27]. These networks/aggregates were broken due to shear stress, and fibrils aligned with the probe at a higher shear. Therefore, at a higher shear, not much difference in the AV was seen, and a similar inconsistency was replicated in the consistency co-efficient and flow behavior index in the case of static heating. Earlier reports of fibrillation of proteins also suggest an increase in AV upon fibrillation for β-Lg [12] and whey proteins [23,28]. 

### 3.3. Protein Oxidation

To understand the protein oxidation, oxidation markers N-formyl kynurenine and Di tyrosine were chosen. Di-tyrosine is mostly used as a marker for protein oxidation, as different oxidative agents can promote its formation [29], while N-Formylkynurenine (NFK) is one of several specific markers for tryptophan oxidation. Also, two marker amino acids, tryptophane and tyrosine, were chosen to see the change in proteins due to heating methods. Figure 5 shows the protein oxidation during the fibrillation of whey proteins under static heating and stirring heating conditions. Figure 5a shows a decrease in tryptophan content with an increase in heating time (0–20 h) in both methods; however, the decrease was higher in the case of static heating compared to stirring heating. Similarly, Figure 5b shows a decrease in tyrosine content with an increase in heating time in both the heating methods, and here also, a decrease in tyrosine was higher in the case of the static heating method. Figure 5c shows an increase in the concentration of tryptophan oxidation compound N-formyl kynurenine with an increase in heating time in both the heating methods, where higher values were seen for the stirring heating method. Figure 5d shows an increase in the concentration of the tyrosine degradation compound Di-tyrosine for both the heating methods; however, the increase in Di-tyrosine value was higher for static heating samples throughout the heating time. These results suggest that static heating is more severe than stirring heating, which leads to higher protein degradation and higher production of oxidation compounds. Keppler et al. [20] studied the oxidation products of tryptophan and tyrosine and reported a decrease in the tryptophan during temperature-induced amyloid aggregation of β-Lg. This decrease in protein or peptide size caused by continuous acid hydrolysis is normal during the fibril formation process as the peptides formed due to hydrolysis are building blocks of fibrils. However, these changes are also affected by oxidative damage of the protein or peptides. Further, it is well known that free-radical mediated oxidation could also induce the cleavage of peptide bonds and result in protein side-chain modifications [30]. Meyer et al. [31] have studied protein oxidation in thermally processed milk and reported that α-La is more likely to be oxidized compared to β-Lg. The current experiment was carried out to identify a specific time after which protein oxidation accelerates. However, the results suggest that protein oxidation continues to proceed throughout the fibrillation period in both methods. Hence, fibrillation time should be minimized to reduce oxidation; however, fibril morphology and maturity should also be taken into consideration. The Th T value and TEM images showed that at 14 h stirring heating can produce more mature fibrils over static heating. 

## 4. Conclusions

The current study aimed to reduce the fibrillation time to make fibrils of milk whey proteins. Based on earlier reported work, the protein concentration (2% *w*/*w*) and pH 2 were finalized. Regarding the time–temperature combination, earlier reported time–temperature combinations were screened in the preliminary study, and a procedure of 80 °C for 20 h was decided for this study. However, a time 20 h is a very long duration. Therefore, the current study aimed to lower the fibrillation time without compromising the structure of fibrils. Based on earlier reports [16], stirring during fibrillation was introduced as a processing intervention. Fibril formation was observed using the Th T value, TEM, which showed fibrils with good morphology can be produced within 14 h with the stirring method. Also, Tricine SDS-PAGE showed that protein was hydrolyzed during the fibrillation process to 2–10 kDa in size and was rearranged into fibrils, which is in line with earlier reported observations. However, peptides derived from proteins other than β-Lg contributing to fibril formation is still open for investigation. Rheological analysis showed that stirring heating produced a more uniform fibril solution with homogeneous viscosity compared to a fibril solution made with static heating. Protein oxidation analysis revealed that prolonged heating increases oxidation products, suggesting reducing fibrillation time as low as possible. Considering all the results, it can be said that heating 2% mWPI solution at pH 2 using stirring heating at 80 °C for 14 h can produce fibrils with good morphology. These findings can be used in further research work on mWPI fibrils and making fibrils at pilot and industrial scales to study the functionality of fibrillated proteins compared to native proteins. 

## Figures and Tables

**Figure 1 foods-13-00466-f001:**
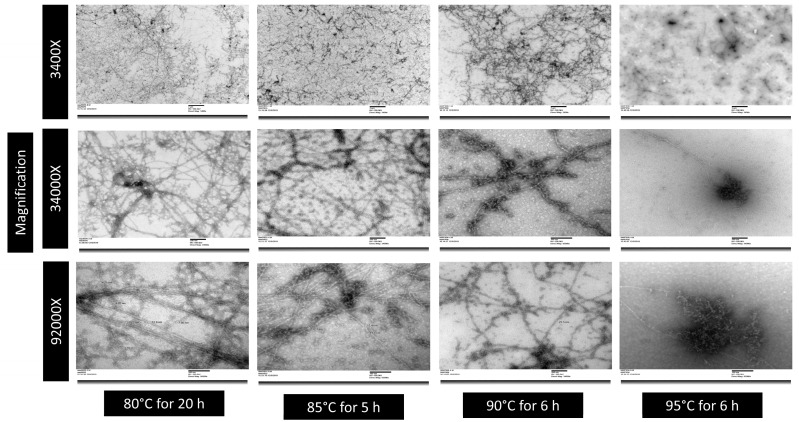
Transmission electron microscopic images of whey protein fibrils produced using 2% mWPI solution (at pH 2) at different time–temperature combinations.

**Figure 2 foods-13-00466-f002:**
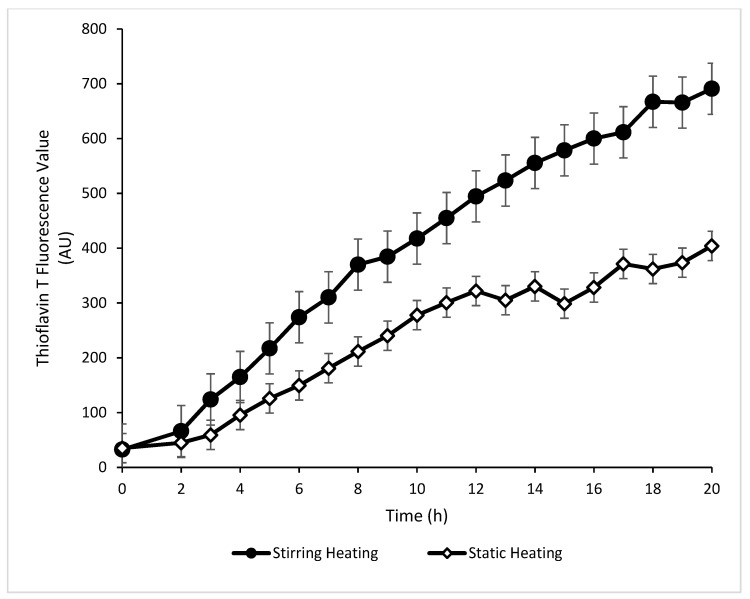
Thioflavin T fluorescence value of mWPI fibril formed under static and stirring heating of mWPI solution (at pH 2) at different time intervals (0–20 h). (Error bars represent standard deviation, n = 2).

**Figure 3 foods-13-00466-f003:**
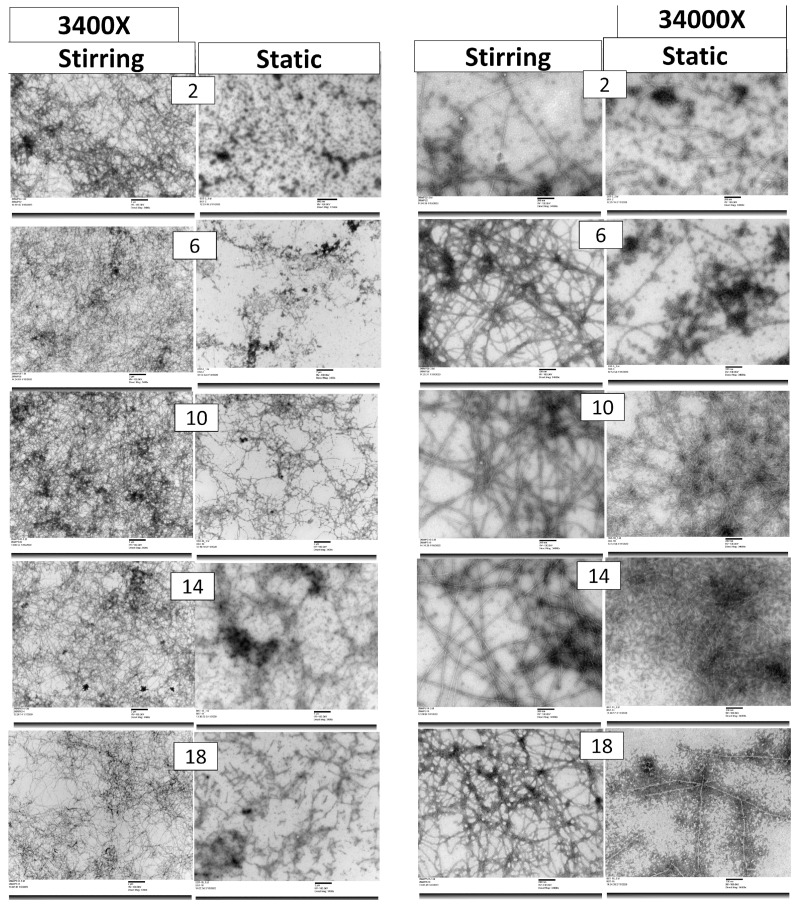
TEM images of fibrils formed under stirring and static heating conditions at different time intervals (2–18 h).

**Figure 4 foods-13-00466-f004:**
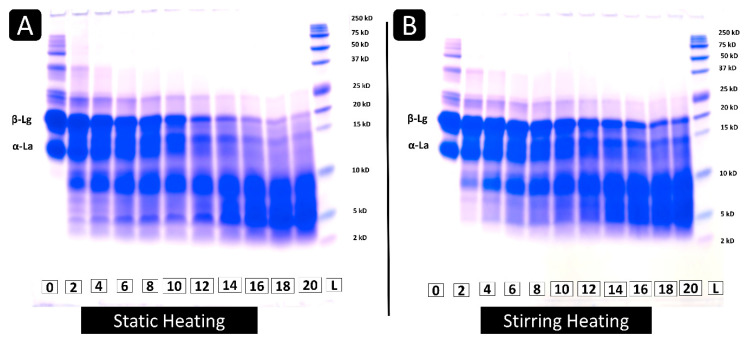
Tricine SDS PAGE of fibril samples formed by static heating (**A**) and stirring heating (**B**) at different time durations (0–20 h, L-ladder).

**Figure 5 foods-13-00466-f005:**
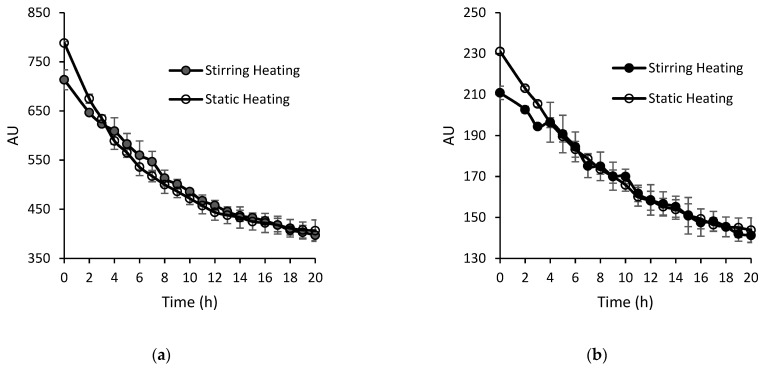
Protein oxidation during stirring and static heating for fibrillation of whey protein. (**a**) Change in the concentration of tryptophan, (**b**) change in the concentration of L-tyrosine, (**c**) change in the concentration of tryptophan oxidation compound N-formyl kynurenine, and (**d**) change in the concentration of L-tyrosine oxidation compound Di-tyrosine. (Error bars represent the standard deviation, n = 2).

**Table 1 foods-13-00466-t001:** Rheology of fibril samples formed by static heating and stirring heating conditions at different time durations (0–20 h).

Time (h)	Apparent Viscosity (m.Pa.s) at 100 s^−1^		Consistency Coefficient (mPa.s^n^)		Flow Behavior Index	
	Stirring Heating	Static Heating	Stirring Heating	Static Heating	Stirring Heating	Static Heating
0	1.08 ± 0.03 ^A^	1.09 ± 0.03 ^A^	1.1 ± 0.2 ^A^	1.2 ± 0.1 ^A^	0.99 ± 0.03 ^A^	0.99 ± 0.01 ^A^
2	1.37 ± 0.19 ^A^	1.16 ± 0.03 ^B^	2.2 ± 0.7 ^A^	1.2 ± 0.1 ^B^	0.91 ± 0.04 ^B^	0.99 ± 0.01 ^A^
3	1.89 ± 0.05 ^A^	1.34 ± 0.07 ^B^	4.6 ± 0.2 ^A^	1.9 ± 0.5 ^B^	0.81 ± 0.01 ^B^	0.94 ± 0.05 ^A^
4	2.23 ± 0.19 ^A^	1.75 ± 0.55 ^A^	6.8 ± 0.9 ^A^	6.3 ± 5.7 ^A^	0.76 ± 0.02 ^A^	0.79 ± 0.14 ^A^
5	2.77 ± 0.31 ^A^	3.58 ± 1.23 ^A^	9.8 ± 1.2 ^A^	38.6 ± 36.4 ^A^	0.72 ± 0.01 ^A^	0.55 ± 0.14 ^B^
6	3.29 ± 0.24 ^A^	4.01 ± 1.55 ^A^	13.3 ± 1.3 ^A^	63.1 ± 62.8 ^A^	0.7 ± 0.01 ^A^	0.49 ± 0.16 ^B^
7	3.68 ± 0.21 ^A^	4.79 ± 1.53 ^A^	16.5 ± 2.2 ^B^	92.2 ± 73.8 ^A^	0.68 ± 0.02 ^A^	0.42 ± 0.12 ^B^
8	4.19 ± 0.39 ^A^	4.21 ± 0.35 ^A^	18.0 ± 2.1 ^B^	58.0 ± 16.4 ^A^	0.68 ± 0.02 ^A^	0.44 ± 0.05 ^B^
9	4.44 ± 0.29 ^A^	4.20 ± 0.56 ^A^	21.9 ± 1.5 ^B^	61.9 ± 21.9 ^A^	0.66 ± 0.01 ^A^	0.43 ± 0.05 ^B^
10	4.72 ± 0.13 ^A^	2.96 ± 1.01 ^B^	22.7 ± 1.4 ^A^	32.8 ± 29.4 ^A^	0.66 ± 0.02 ^A^	0.54 ± 0.12 ^B^
11	4.92 ± 0.16 ^A^	5.22 ± 1.56 ^A^	24.6 ± 0.9 ^B^	101.1 ± 64.4 ^A^	0.65 ± 0.02 ^A^	0.4 ± 0.09 ^B^
12	5.20 ± 0.14 ^A^	3.69 ± 1.21 ^A^	27.9 ± 2.9 ^A^	42.8 ± 33.1 ^A^	0.64 ± 0.02 ^A^	0.51 ± 0.1 ^B^
13	5.27 ± 0.16 ^A^	5.63 ± 1.63 ^A^	27.2 ± 1.1 ^B^	133.1 ± 49.0 ^A^	0.65 ± 0.02 ^A^	0.33 ± 0.03 ^B^
14	5.31 ± 0.07 ^A^	5.01 ± 0.91 ^A^	27.7 ± 1.0 ^B^	108.9 ± 32.0 ^A^	0.64 ± 0.01 ^A^	0.35 ± 0.06 ^B^
15	5.49 ± 0.06 ^A^	5.64 ± 2.58 ^A^	29.4 ± 2.0 ^A^	1564.2 ± 2.9 ^A^	0.64 ± 0.02 ^A^	0.41 ± 0.17 ^A^
16	5.06 ± 0.90 ^B^	5.62 ± 1.10 ^A^	28.8 ± 6.8 ^B^	142.6 ± 0.0 ^A^	0.63 ± 0.02 ^A^	0.3 ± 0.07 ^B^
17	6.09 ± 0.14 ^A^	3.84 ± 0.83 ^B^	33.7 ± 3.8 ^A^	68.7 ± 0.1 ^A^	0.63 ± 0.02 ^A^	0.41 ± 0.11 ^B^
18	6.07 ± 0.11 ^A^	6.48 ± 1.23 ^A^	36.8 ± 0.6 ^B^	241.8 ± 0.1 ^A^	0.61 ± 0.01 ^A^	0.23 ± 0.08 ^B^
19	6.08 ± 0.09 ^A^	6.48 ± 1.10 ^A^	33.0 ± 2.3 ^B^	161.1 ± 0.0 ^A^	0.63 ± 0.02 ^A^	0.31 ± 0.04 ^B^
20	5.94 ± 0.13 ^A^	6.06 ± 1.01 ^A^	32.3 ± 4.0 ^B^	128.7 ± 0.0 ^A^	0.64 ± 0.02 ^A^	0.35 ± 0.06 ^B^

^A,B^ Means within a row (static heating and stirring heating) with different superscripts differ (*p* < 0.05) (n = 2).

## Data Availability

Data is contained within the article.

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
