# Peer review of "Milk Whey Protein Fibrils—Effect of Stirring and Heating Time"

_foods, 2024, doi:10.3390/foods13030466_

Round 1
Reviewer 1 Report
Comments and Suggestions for Authors
Title: Milk whey protein fibrils – effect of stirring and heating time
Manuscript ID: foods-2798020
Type of manuscript: Article
General comment:
The subject of the manuscript is relevant and has significance in advancing knowledge. The manuscript is well-written and presented. However, a few questions need to be clarified to improve the quality of the manuscript.
Title: Why Authors are writing Milk whey proteins fibrils. Whey protein fibrils should be sufficient. If this is native whey protein directly from milk, then needs to be addressed.
Line 58: mWPI- same comment as above.
Line 88: “was” should be “is”.
Line 99: Approach 2: How much sample was taken for heating? Also, the authors withdrew 10 mL samples every 2 hours and then how the heat capacity was maintained- for example, if 150 mL was the initial volume after 5 samples 50 mL (10 mL/sample) was withdrawn- would the remaining volume of 100 mL change the heat capacity and functionality or fibril formation denaturation and other properties. Similarly, after the last sample 60 ml will be left- will these conditions affect the results? Please explain.
Line 238-242: the sentence is too long and difficult to read. Please consider it breaking into smaller and easy-to-understand sentences.
Line 256: are these 25 kDa casein fragments or heat-denatured proteins while processing mWPI? And whether they act as nuclei.
Line 268: why the 25 kDa hydrolyzed? A better explanation will be helpful.
Table 1: the formatting of the consistency coefficient can be better.
Author Response
Reviewer 1:
General comment:
The subject of the manuscript is relevant and has significance in advancing knowledge. The manuscript is well-written and presented. However, a few questions need to be clarified to improve the quality of the manuscript.
AU: Thank you.
Title: Why Authors are writing Milk whey proteins fibrils. Whey protein fibrils should be sufficient. If this is native whey protein directly from milk, then needs to be addressed.
AU: Most of the studies so far used whey protein isolates fractionated from cheese whey. Consequently, WPI derived from cheese whey contains GMP along with other major whey proteins. This study used whey protein isolate derived directly from skim milk and does not contain GMP. In order to differentiate these two WPIs, we followed ADPI nomenclature in our previously published manuscripts. This also ensures continuity and adds value to the current manuscript.
- Rathod, G., & Amamcharla, J. K. (2021). Process development for a novel milk protein concentrate with whey proteins as fibrils. Journal of Dairy Science, 104(4), 4094–4107. https://doi.org/10.3168/JDS.2020-19409
- Rathod, Gunvantsinh, Boyle, D. L., & Amamcharla, J. K. (2022). Acid gelation properties of fibrillated model milk protein concentrate dispersions. Journal of Dairy Science, 105(6), 4925-4937.
Line 58: mWPI- same comment as above.
AU: mWPI to differentiate whey protein from cheese whey. Explained in earlier response.
Line 88: “was” should be “is”.
AU: Modified as suggested.
Line 99: Approach 2: How much sample was taken for heating? Also, the authors withdrew 10 mL samples every 2 hours and then how the heat capacity was maintained- for example, if 150 mL was the initial volume after 5 samples 50 mL (10 mL/sample) was withdrawn- would the remaining volume of 100 mL change the heat capacity and functionality or fibril formation denaturation and other properties. Similarly, after the last sample 60 ml will be left- will these conditions affect the results? Please explain.
AU:This is a valid point. However, heat capacity is an extensive property of the system and can change based on the volume/weight of the system. However, temperature and concentration are intensive properties and are independent of the size of the system. Fibril formation is temperature dependent rather than heat capacity. We also considered the effect of volume reduction during the course of the experiments and can change the mixing characteristics of the system especially for stirring experiments. To minimize this effect, we started with 500 mL sample and used only 200 mL for sampleing at regular intervals. At the end of study, we still had 300mL of unused sample.
Line 238-242: the sentence is too long and difficult to read. Please consider it breaking into smaller and easy-to-understand sentences.
AU: Sentence was revised. Line no: 256-261.
Line 256: are these 25 kDa casein fragments or heat-denatured proteins while processing mWPI? And whether they act as nuclei.
AU: They are casein fragments most likely beta-casein which pass through the membrane during cold microfiltration. The mWPI produced using microfiltration may contain some caseins as membrane processing is not as selective as enzymatic action (rennet). There are minor whey proteins such as Bovine serum albumin, and immunoglobuline which have higher molecular weight. They may not act as nuclei, for fibrils smaller molecular weight peptides are needed.
Line 268: why the 25 kDa hydrolyzed? A better explanation will be helpful.
AU: Under high heat and low pH all protein fractions were hydrolyzed to smaller molecular weight peptide. Line no 286-288 improved.
Table 1: the formatting of the consistency coefficient can be better.
AU: Consistency coefficient column formated
Reviewer 2 Report
Comments and Suggestions for Authors
Main Impression
The study and knowledge adressed in this manuscript is important to the field of valorization of whey. The work is an important contribution to processoptimation of fibril-formation of whey, as these results can improve the functional propertis of whey based ingredients. The article fits into the scope of the journal, with its focus on different aspects of food technology, processing and application. Yet, more documentation is needed to reach the industrial relevance.
Overall, the language in the manuscript is good and easy to read, but some phrases can lead to misunderstanding. These are pointed out in the detailed report. Detailed comments and suggestions for improvement are given below.
No acknowledgement from funding or coworkers are given. Please make sure to credit all contributors.
ABSTRACT
The abstract neatly gives the key to the article, and is well written.
L10-11 If milk whey proteins are the raw material used in this study, then mWPI, the abbreviation, needs to be inserted in the text.
L13 What challenges need to be overcome?
L21 Is this correct. Is it actually static heating that will help upscaling?
INTRODUCTION
The introduction gives a good background and knowledge base on previous work performed with fibril formation from whey. The introduction also clearly shows the need for more detailed information on the processing and application of native whey, especially whey approaching an industrial scale. Are there any industrial production of fibrils today?
Preferably more focus could be put on the optimal fibrils, e.g. explain the term “mature fibril”, and indicate what is the optimum length, diameter and uniformity of fibrils. And how does the fibril character affect their functionality and application as ingredients? Also, it is slightly confusing what type of whey is used for the different studies. Please clearify how the choice of whey (cheese whey, milk whey or pure bLg-fraction) affect the fibril-formation. What is the optimal raw-material, and why?
The introduction lacks a clear purpose of the present work, and how this fits into the previos work in the area. Please include the aim of the study in the introduction.
MATERIALS AND METHODS
Number of replicate measurements are completely missing from all the methods described. Including the experimental setup from two lots of mWPI. Please fulfill M&M with this information for all methods and designs.
L82-84 Two lots of mWPI are used. How many productions of fibrils are prepared from each lot?
L91-94 Does this mean that the mWPI solutions were kept cold over two nights?
L95-96 Does the size of the samples make the results relevant for industrial applications – or would the experiment need to be repeated in different scales? If yes, please insert a comment in the end of the discussion about future work.
L100 Please include the size of the glass container.
L112-115 Please insert number of replicates
L117-118 Please insert number of replicates
L128-129 How many pictures per replicate?
L141-144 Please insert number of replicates
RESULTS
Results are presented and discussed in the same paragraph. It is suggested to rewrite this part of the paper to Results AND DISCUSSION.
The content in this chapter is grouped after analytical method, thus the story of fibrilliation and ingredient-functionality is not very clear.
L153 Rename the chapter, Results and Discussion
L158-159 The samples are unheated, making the start-reference. Rephrase to visualize that these samples are unheated.
L198 To interpret the findings in Figure 2, we may say that stirring gives faster? or larger? or both? formation of fibrils. Please insert a short comment on the overall interpretation of the figure, after the figure is presented in the text.
L248 Please interpret Figure 3 with a short comment after the figure. How can we see that 14h is the optimal time, and how can we see the uniformity of the fibrils in this figure?
L276 Please doublecheck if the sentence is ok. Is it stirring or static heating that showed lower conversion of small-sized protein fractions.
L284-286 A statement that needs to be included in the conclusion when listing future work?
L331-332 An increase in AV is expected after fibrillation, but what is actually the wanted effect? Lower AV during production? Please identify what is optimal results from these measurements. Also remember to include number of replicates.
L356-358 Is this in line with the results in Figure 4?
DISCUSSION
The discussion is written in the form of a conlcusion. The conclusion neatly sums up the findings of this work. Please include more comments on what is needed in future research.
L 379 Rewrite to Conclusion
L396-398 Please include more information about this in the introduction og this paper (functionality of fibrils vs native proteins).
TABLES AND FIGURES
Throughout the article the results from the figures are repeated in the text. Reconsider it this is necessary.
Figure 2. Explain the errorbars (how many replicates are used to calculate the standarddeviation)?
Table 1. Reconsider the presentation of these results. The table is not very easy to read, and the significant results could be presented easier in a figure, prehaps? Also rewrite the footnote under the table, what is actually compared in the statistical analysis?? Remember to include number of replicates
Figure 5. Is it correct to say that figur a) and figur c) should be read together, and figure b) together with figure d)? Please name the individual figures. Please explain the errorbars, and identify significant results – this is especially important since the y-axis are of very different scales in figures a-d. Remember to include number of replicates.
Author Response
Main Impression
The study and knowledge adressed in this manuscript is important to the field of valorization of whey. The work is an important contribution to processoptimation of fibril-formation of whey, as these results can improve the functional propertis of whey based ingredients. The article fits into the scope of the journal, with its focus on different aspects of food technology, processing and application. Yet, more documentation is needed to reach the industrial relevance.
AU: Thank you
Overall, the language in the manuscript is good and easy to read, but some phrases can lead to misunderstanding. These are pointed out in the detailed report. Detailed comments and suggestions for improvement are given below.
No acknowledgement from funding or coworkers are given. Please make sure to credit all contributors.
AU: Given in Line no-427-432.
ABSTRACT
The abstract neatly gives the key to the article, and is well written.
L10-11 If milk whey proteins are the raw material used in this study, then mWPI, the abbreviation, needs to be inserted in the text.
AU: mWPI was inserted in Line 15- for milk whey protein isolate.
L13 What challenges need to be overcome?
AU: Text was added- Line no-13-14.
L21 Is this correct. Is it actually static heating that will help upscaling?
AU: It is stirring heating. Corrected in text line no: 22.
INTRODUCTION
The introduction gives a good background and knowledge base on previous work performed with fibril formation from whey. The introduction also clearly shows the need for more detailed information on the processing and application of native whey, especially whey approaching an industrial scale. Are there any industrial production of fibrils today?
AU: We did not come across any commercial industrial-scale production of fibrils. Whey protein fibrils as a sole ingredient is difficult, commercially not available to our knowledge.
Preferably more focus could be put on the optimal fibrils, e.g. explain the term “mature fibril”, and indicate what is the optimum length, diameter and uniformity of fibrils. And how does the fibril character affect their functionality and application as ingredients? Also, it is slightly confusing what type of whey is used for the different studies. Please clearify how the choice of whey (cheese whey, milk whey or pure bLg-fraction) affect the fibril-formation. What is the optimal raw-material, and why?
AU: Mature fibrils-Details were added- Line-50-53, 57-59. Fibrils structure and functionality- Line no-34-37. mWPI preference- Line no-62-65.
The introduction lacks a clear purpose of the present work, and how this fits into the previos work in the area. Please include the aim of the study in the introduction.
AU: Purpose of present work- Line no 78-83, Previous work- Line no- 44-50, 55-59, Preliminary study to fit/align previous work Line no-59-61, Aim- Line no-78-83.
MATERIALS AND METHODS
Number of replicate measurements are completely missing from all the methods described. Including the experimental setup from two lots of mWPI. Please fulfill M&M with this information for all methods and designs.
AU: Mentioned in line no: 161 under 2.6 statistical analysis.
L82-84 Two lots of mWPI are used. How many productions of fibrils are prepared from each lot?
AU: Mentioned in line no: 90 as suggested.
L91-94 Does this mean that the mWPI solutions were kept cold over two nights?
AU: Yes, first overnight hydration, second overnight after pH adjustment to 2. Microbial growth is not a concern at pH 2. Fibrillation started 3rd day early morning.
L95-96 Does the size of the samples make the results relevant for industrial applications – or would the experiment need to be repeated in different scales? If yes, please insert a comment in the end of the discussion about future work.
AU: This size of the experimental setup is a lab-scale version of process control, which we can perform at an industrial scale. We performed fibrillation at the pilot scale, following the lab scale setup. Comment added Line no: 417-419
L100 Please include the size of the glass container.
AU: Added in line no:107.
L112-115 Please insert number of replicates.
AU: Inserted at corrosponding analysis method. Line no-124.
L117-118 Please insert number of replicates
AU: Inserted line no: 117.
L128-129 How many pictures per replicate?
AU: 5 five pictures were taken at different scale, shown only 3400X and 34000X.
L141-144 Please insert number of replicates
AU: Mentioned in Line no: 158.
RESULTS
Results are presented and discussed in the same paragraph. It is suggested to rewrite this part of the paper to Results AND DISCUSSION.
AU: Rewritten as result and discussion, Line no: 164.
The content in this chapter is grouped after analytical method, thus the story of fibrilliation and ingredient-functionality is not very clear.
AU: The story is based on three aspects, confirmation of fibril formation through ThT, TEM, and Tricine SDS, then functionality with their viscosity, and protein oxidation as impact on proteins. Based on these parameters, the fibrillation method and time are suggested.
L153 Rename the chapter, Results and Discussion
AU: Renamed as “Results and Discussion”- Line no: 164.
L158-159 The samples are unheated, making the start-reference. Rephrase to visualize that these samples are unheated.
AU: Information added- Line no-168-170.
L198 To interpret the findings in Figure 2, we may say that stirring gives faster? or larger? or both? formation of fibrils. Please insert a short comment on the overall interpretation of the figure, after the figure is presented in the text.
AU: Text added- Line no 178-179
L248 Please interpret Figure 3 with a short comment after the figure. How can we see that 14h is the optimal time, and how can we see the uniformity of the fibrils in this figure?
AU: Text inserted- Line no 215-221 and Line no-232-234.
L276 Please doublecheck if the sentence is ok. Is it stirring or static heating that showed lower conversion of small-sized protein fractions.
AU: The sentence is okay. Stirring heating showed lower conversion.
L284-286 A statement that needs to be included in the conclusion when listing future work?
AU: Added in conclusion- Line 410-411.
L331-332 An increase in AV is expected after fibrillation, but what is actually the wanted effect? Lower AV during production? Please identify what is optimal results from these measurements. Also remember to include number of replicates.
AU: An increase in AV is expected after fibrillation compared to the unfibrillated/control sample. An increase in AV shows thickening functionality improvement. Higher AV is desirable after fibrillation, but we did not find literature on the optimum number as it may vary with source, protein concentration, and fibrillation process. Number of replicates added in Method-Line no: 144-145.
L356-358 Is this in line with the results in Figure 4?
AU: Yes, the results are in line with Figure 4.
DISCUSSION
The discussion is written in the form of a conlcusion. The conclusion neatly sums up the findings of this work. Please include more comments on what is needed in future research.
L 379 Rewrite to Conclusion
AU: Renamed to conclusion. Line no- 399.
L396-398 Please include more information about this in the introduction og this paper (functionality of fibrils vs native proteins).
AU: Included Line no: 28-34
TABLES AND FIGURES
Throughout the article the results from the figures are repeated in the text. Reconsider it this is necessary.
AU: Key results are reported in the text for easy understanding of the manuscript.
Figure 2. Explain the error bars (how many replicates are used to calculate the standard deviation)?
AU: 2 Replicates were used.
Table 1. Reconsider the presentation of these results. The table is not very easy to read, and the significant results could be presented easier in a figure, prehaps? Also rewrite the footnote under the table, what is actually compared in the statistical analysis?? Remember to include number of replicates
AU: Mentioned in line 355-356.
Figure 5. Is it correct to say that figure a) and figure c) should be read together, and figure b) together with figure d)? Please name the individual figures. Please explain the error bars, and identify significant results – this is especially important since the y-axis are of very different scales in figures a-d. Remember to include number of replicates.
AU: Yes, it can be compared in said way. However, we reported a decrease in both amino acids first, then moved to oxidation products. Figures were prepared following journal format (therefore names are given in the figure caption. Description of error bar added- Line no- 397-398.
Reviewer 3 Report
Comments and Suggestions for Authors
Line 69-76: The purpose should be better explained
Line 91-109: References are missing
Chapter 3.3.: can this be explained any better?
Chapter 4: This chapter should be moved to chapter 3. Because in this form it is too short and duplicates already cited literature.
I suggest entering 2-3 summary sentences as Conclusions.
Comments on the Quality of English LanguageMinor editing of English language required
Author Response
Line 69-76: The purpose should be better explained
AU: Text improved Line no- 78-83.
Line 91-109: References are missing
AU: Reference added Line no: 98. The method is an extension of our earlier research work.
Chapter 3.3.: can this be explained any better?:
AU: We tried as per our understanding.
Chapter 4: This chapter should be moved to chapter 3. Because in this form it is too short and duplicates already cited literature.
AU: Chapter 4 is renamed as a conclusion to summarize, as per suggestion.
I suggest entering 2-3 summary sentences as Conclusions.
AU: Chapter 4 was presented as a conclusion. Added required information for clarity. Line no-400-419.